# Chance elections, social distancing restrictions, and KENTUCKY's early COVID-19 experience

Charles Courtemanche[1]*, Joseph Garuccio[2], Anh Le[1], Joshua Pinkston[1,3], Aaron Yelowitz[1]

1 Department of Economics, Gatton College of Business and Economics, University of Kentucky, Lexington, Kentucky, United States of America, 2 Department of Economics, Andrew Young School of Policy Studies, Georgia State University, Atlanta, Georgia, United States of America, 3 Department of Economics, College of Business, University of Louisville, Louisville, Kentucky, United States of America

These authors contributed equally to this work.
* courtemanche@uky.edu

**Data Availability Statement:** The data underlying this study are publicly available at: https://www.openicpsr.org/openicpsr/project/140481/version/V1/view

## Abstract

Early in the pandemic, slowing the spread of novel coronavirus disease 2019 (COVID-19) relied on non-pharmaceutical interventions. All U.S. states adopted social-distancing restrictions in March and April of 2020, though policies varied both in timing and scope. Compared to states with Democratic governors, those with Republican governors often adopted measures for shorter durations and with greater resistance from their residents. In Kentucky, an extremely close gubernatorial election immediately prior to the discovery of SARS-CoV-2 replaced a Republican incumbent with a Democrat, despite Republicans easily winning all other statewide races. This chance election result offers a unique opportunity to examine the impact of early social distancing policies in a relatively conservative, rural, white-working-class state. Our study begins by estimating an event-study model to link adoption of several common social distancing measures–public school closures, bans on large gatherings, closures of entertainment-related businesses such as restaurants, and shelter-in-place orders (SIPOs)–to the growth rate of cases across counties in the Midwest and South in the early stages of the pandemic. These policies combined to slow the daily growth rate of COVID-19 cases by 9 percentage points after 16 days, with SIPOs and entertainment establishment closures accounting for the entire effect. In order to obtain results with more direct applicability to Kentucky, we then estimate a model that interacts the policy variables with a "white working class" index characterized by political conservatism, rurality, and high percentages of white, evangelical Christian residents without college degrees. We find that the effectiveness of early social distancing measures decreased with higher values of this index. The results imply that the restrictions combined to slow the spread of COVID-19 by 12 percentage points per day in Kentucky's two largest urban counties but had no statistically detectable effect across the rest of the state.

**Funding:** The authors received no specific funding for this work.

**Competing interests:** The authors have declared that no competing interests exist.

## 1. Introduction

The COVID-19 pandemic has become not only the defining public health challenge of our time but also the greatest economic threat since the Great Recession. As cases began to emerge in the United States in March 2020, states and localities implemented social distancing restrictions that varied in timing and scope. For instance, in Kentucky, the first major interventions took effect on March 16, when Governor Andy Beshear closed public schools and the dining-in sections of restaurants and bars [1]. Governor Beshear soon closed other non-essential businesses, banned large gatherings, and ultimately issued a "Healthy at Home" directive that was functionally equivalent to the shelter-in-place orders (SIPOs) enacted in most states [2, 3].

Epidemiological models link the frequency and nature of social interactions to case growth but tend to simply assume that particular government restrictions reduce these interactions to a given level. The well-known Imperial College London and University of Washington models project case trajectories with social distancing versus no social distancing, but they are agnostic about how the specified level of social distancing would be achieved [4] [5]. Economists' policy evaluation toolkit can therefore help fill a critical void in the literature.

Accordingly, recent studies provide econometric evidence on the effectiveness of these social-distancing measures in the U.S. Several studies use publicly available mobility data from cell phones to document the extent to which social distancing restrictions influenced movement [6–10]. Others examine impacts on the spread of COVID-19 more directly. For instance, one paper uses synthetic control methods to show that California's SIPO reduced COVID-19 case growth in the first three weeks following its implementation [11]. Our previous work estimates an event-study model with all U.S. counties and show that SIPOs strongly reduced the growth rate of COVID-19 cases, while closing entertainment-related businesses (including bars and restaurants) had a moderate effect [12]. However, they find no evidence that closing public schools or banning large gatherings reduced case growth without a broader SIPO also being in place.

While social distancing policies may have slowed the spread of COVID-19 in California and the U.S. as a whole, there is a clear concern with whether a "one-size-fits-all" policy approach is sustainable or effective given underlying differences in politics, density, urbanicity, and attitudes towards coronavirus. Evidence suggests that SIPOs were most effective in early adopting states and those with high population densities [13]. Other work finds significant differences in the efficacy of SIPOs in Texas, with far greater effectiveness in early-adopting urbanized counties relative to late-adopting counties induced by the statewide mandate [14]. Broadly speaking, the pattern found in Texas early in the pandemic is mirrored throughout the country: policy choices by governors or local officials tend to reflect a host of factors–including underlying attitudes of their constituents–regardless of the spread of the virus. Such attitudes led to the Sturgis motorcycle rally in South Dakota becoming a likely super-spreader event [15].

Because of the difficulty in separating policy choices from underlying attitudes about the virus, few studies have presented convincing counterfactual examples. One notable exception relating to the reopening of the economy was a natural experiment in which the Wisconsin Supreme Court abolished the state's "Safer at Home" order in May 2020, an action which was largely unanticipated and did not appear to increase the spread of the virus [16]. Another example–from England–relates to contract tracing effectiveness where follow-up on some COVID-19 cases was delayed due to an Excel spreadsheet truncating cases, resulting in substantially more cases and deaths [17].

In Kentucky's 2019 gubernatorial race, a Democratic challenger, Andy Beshear, unseated the Republican incumbent, Matt Bevin, in a close and arguably surprising election result.

Governor Beshear won by 5,136 votes, which translates to a margin of less than 0.4 percentage points. In contrast, all of the other five statewide races in November 2019 went to the Republican candidate by margins of 5 to 21 percentage points [18]. In both 2016 and 2020, former President Trump won Kentucky with over 62% of the votes and margins of 25 to 30 percentage points. Bevin was widely considered an unusually divisive and inflammatory incumbent, which manifested in consistently low approval ratings. For example, one political scientist argued he "committed a series of unforced errors in his battle with teachers' unions, using much more provocative language than necessary, much more combative language than necessary." In the summer of 2019, Bevin had the lowest approval rating of any governor in the nation, and a 40% disapproval rating among Kentucky Republicans [19].

Additionally, Gov. Beshear–as the son of a former governor–had a name recognition advantage relative to the typical Kentucky Democrat [20]. The combination of these factors led to a close election in a typically reliably red state.

As a consequence of this election result, Kentucky's policy choices several months later–the result of broad emergency powers allowed to the governor–represent essentially a coin-flip between more stringent lockdown measures often instilled by Democratic governors and more lax measures by Republican governors. In 2021, the Kentucky legislature–with Republican super-majorities–passed bills aimed at limiting the power of the state governor [21]. Even with narrow victory margins, elected officials tend to not converge on policies [22], suggesting that Kentucky voters unwittingly elected a more stringent set of social distancing policies shortly before the pandemic. Although Bevin's policy responses to the pandemic were obviously not observed, on March 11, 2020 –the day the World Health Organization declared COVID-19 a pandemic–he openly mocked Beshear's early response in a tweet that was widely criticized for not taking the situation seriously. The tweet read, "BREAKING NEWS: Chicken Little has just confirmed that the sky IS indeed falling. . . Everyone is advised to take cover immediately and to bring lots of toilet paper with them when they do so. . .". At the time, Kentucky had eight confirmed cases, and Beshear was taking actions to protect nursing home residents.

Examining Kentucky is also insightful because it provides a window into the COVID-19 experience of a predominantly white, working class state. Compared to the U.S. as a whole, Kentucky is considerably more rural (population-weighted population density of 555 people per square mile compared to 2,930) and has a much higher share of residents who are non-Hispanic white (84% versus 61%), lack a college degree (76% versus 68%), are Evangelical Christians (33% versus 16%), and voted for President Trump (63% versus 46% in 2016).

It is unclear the extent to which prior studies on California, Texas, Wisconsin, or the U.S. as a whole are applicable to states like Kentucky. For instance, SIPOs are difficult to enforce and many (including Kentucky's) have not specified formal penalties for violations [23]. SIPOs may therefore be best characterized as a "nudge" with social pressure and information about the severity of the pandemic–rather than threat of legal penalties–being the main channels through which they operate [24]. Viewed through that lens, it would not be surprising if SIPOs were less effective in areas where residents' political views or other attributes led to skepticism of their appropriateness [25]. The effectiveness of the more readily enforceable social distancing restrictions, such as school and business closures and bans on large gatherings, could also vary. Less affluent, sparsely populated communities tend to have fewer restaurants and other businesses as well as fewer major events than larger, more affluent urban centers, so one might expect residents' daily lives to be less affected by their closure. Additionally, what residents do instead after the closures could vary by education, political views, race, ethnicity, and religion. For instance, once restaurants closed, some people who might otherwise have gone out to eat would have stayed home alone, others would have gathered together at a private residence,

and still others might have gathered for an outdoor picnic. Similarly, kids attending closed schools could have sheltered in place at home, gone to whatever public places were still open, or congregated at friends' houses. Such responses to restrictions likely varied across numerous cultural and socioeconomic dimensions.

We contribute to the rapidly growing literature on the effectiveness of social distancing restrictions by aiming to better understand their impacts in the South and Midwest in general, and Kentucky in particular. We begin by estimating an event-study model similar to our previous work using the whole U.S. [12], but with the sample restricted to counties in Southern and Midwestern states. The results largely mirror those found in our prior work despite the narrower geography [12]. SIPOs were the most effective policy in slowing the spread of COVID-19; closing entertainment-related facilities such as restaurants, bars, and gyms was the second most effective; and closing schools and banning large events (in the absence of a SIPO) had no statistically discernable effect. In all, these policies combined to reduce the daily growth rate of COVID-19 cases by 9 percentage points–a substantial impact relative to the sample mean of 11%.

Next, realizing that substantial heterogeneity exists even within the South and Midwest, we estimate a model that interacts the policy variables with a summary index characterizing "white, working-class" counties (based on race/ethnicity, population density, education, Trump vote share, and religion). Not surprisingly in light of the discussion above, we find that the social distancing restrictions were less effective in slowing the spread of COVID-19 in counties with higher values of this index. The results imply that the restrictions combined to slow the spread of COVID-19 by 12 percentage points per day in Kentucky's two largest urban counties–Jefferson (Louisville) and Fayette (Lexington)–but had no statistically detectable effect across the rest of the state.

## 2. Data

Our dataset includes daily official case counts from each of the 2,477 counties (or equivalents such as parishes and independent cities) in the South and Midwest. Following our previous work [12], our sample period starts on March 5, 2020, the day the U.S. reached 100 total cases (and the starting point on many graphs in the popular press of COVID-19 case growth). This was also the day before Kentucky's first official case. Our sample period ends on April 25 to coincide with when states began removing restrictions. The resulting sample size is 128,388 county-by-day observations. In all analyses, each county observation is weighted by population using 2018 estimates [26].

Our COVID-19 case data come from the Johns Hopkins Center for Systems Science and Engineering's 2019 Novel Coronavirus COVID-19 Data Repository, which includes information from an array of sources such as government and independent health institutions [27]. We use these data to compute each county's daily exponential growth rate in confirmed COVID-19 cases, which is equal to the natural log of daily COVID-19 cases minus the log of daily COVID-19 cases on the prior day, multiplied by 100 to be interpretable as percent. Following a prior study [28], we add one to the case totals to prevent the log of cases from being undefined when there are no cases in a county on a given day. The sample mean is 507 cases, so adding one is only a small distortion that does not materially impact the results. The growth rate was multiplied by 100 and can be read as percentage point changes.

Although the confirmed case counts that we use to create growth rates understate true prevalence [29], they were a critical metric in the Trump administration's "Opening Up America Again" plan [30]. The number of cases and incidence (daily new cases per 100,000) continue to be key metrics in guidance from the CDC, state governments and the White House Coronavirus Task Force.

We make use of five county-level characteristics to construct what we call, for lack of a better term, our "white-working-class" index. These characteristics include the percent of the population that is non-Hispanic white, percent of adults without a bachelor's degree, population sparsity (negative of density), percent evangelical, and the Trump vote share in the 2016 presidential election. In each case, higher values of each of these five variables lead to higher values for the summary index. The data for the first three factors are sourced or constructed from data provided by a team at Johns Hopkins [31]. The data on evangelical populations come from the U.S. Religion Census [32] and the vote share data come from the MIT Election Data and Science Lab and the Harvard Dataverse. To construct the index, we follow a procedure developed by prior literature that is now common in economics [33]. We first standardize each characteristic variable by subtracting its mean and dividing by its standard deviation. Then, we sum the five standardized variables and divide by the standard deviation of this sum. The result is a composite measure with a standard deviation of one and higher values unambiguously pointing in the direction of the county being more heavily "white working class".

As discussed in the introduction, this index is useful to analyze Kentucky because the state has particularly high values of each of these variables compared to the U.S. as a whole. In fact, the state is even relatively heavily "white working class" within our restricted sample of Midwestern and Southern counties. After weighting the counties by population to determine state averages, Kentucky's percent white is 84 compared to 63 for the full Midwest and South, its percent without a bachelor's degree is 76 compared to 70, its Trump vote share is 62 compared to 50, its percent evangelical is 33 percent compared to 22, and its population-weighted population density is roughly half. Kentucky's value of the composite index is 0.94 –meaning it is nearly a full standard deviation more "white working class" than the Midwest and South as a whole.

Our information on state and local government social-distancing interventions comes from our previous work [12], which is itself largely drawn from work by the aforementioned team at Johns Hopkins [31], with a few errors corrected. The majority of the policy variation, including all variation in Kentucky, is at the state level, but a number of counties implemented restrictions prior to the state, and our use of county-level data allows these restrictions to be incorporated.

As in our previous work [12], we study four separate interventions. One is whether public schools were closed, with the closure coded as taking effect on the first cancelled school day. This was March 16 in Kentucky. Another is whether there was a closure of at least some entertainment-related businesses. Formally, this is an indicator for either restaurant dining areas (including bars) or gyms/entertainment centers being closed. In practice, the two types of closures were almost always enacted at either the same time or close to it. In Kentucky, restaurants were closed on March 16 and gyms, entertainment centers, and other businesses that involve gathering were closed just two days later; this variable is therefore set to one starting on March 16. The third intervention is a ban on large gatherings that is at least as restrictive as 500 or more people. Most such bans, including Kentucky's (which took effect on March 20), applied to gatherings of 50 or more people. Finally, the strongest restriction is the SIPO. The team at Johns Hopkins coded Kentucky's SIPO as starting on March 26 [31]. While Governor Beshear has not officially used the term SIPO, March 26 is the date he closed all non-life-sustaining business, which made Kentucky's "Healthy at Home" initiative functionally equivalent to the SIPOs in place in other states.

Our sample period begins on March 5, 2020, which is the day before Kentucky reported its first case. The number of COVID-19 cases in Kentucky grew to 3,857 by April 25, the last day in our sample. Fig 1 illustrates the reach of social-distancing policies on the population in the Midwest and South over time relative to the timing of Kentucky in implementing these

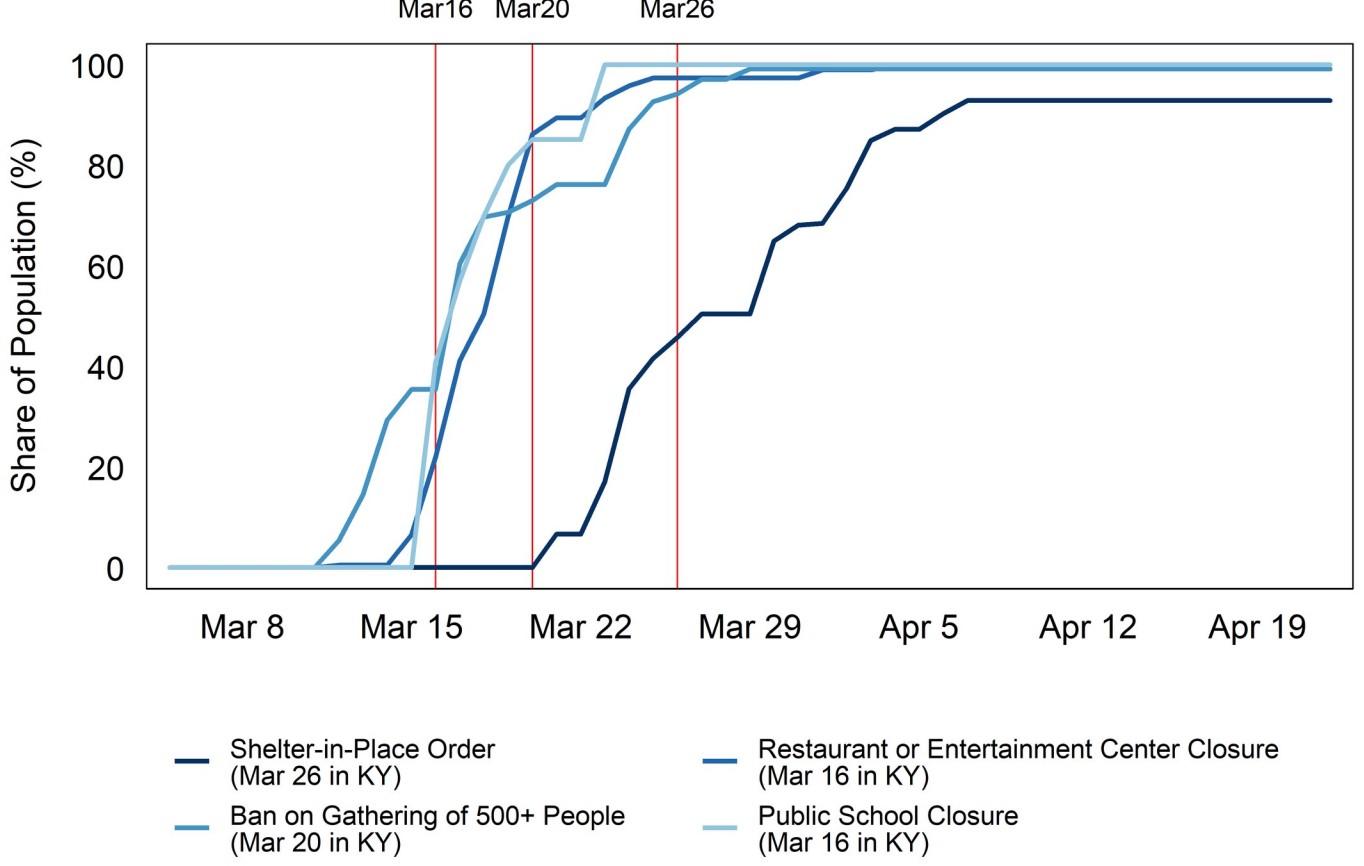

**Fig 1. Fraction of population in Southern and Midwestern states covered by social-distancing measures.** Notes: Authors' calculations from population-weighted county data. The red lines indicate Kentucky's adoption dates.

measures (indicated by the vertical red lines). 22 percent of the population in the South and Midwest lived in counties where schools were already closed when Kentucky's closure took effect. 41 percent of the population in these regions lived in counties with restaurant or entertainment center closures when Kentucky did so. When Kentucky officially banned large gatherings, similar orders already affected 73 percent of people in these regions. About 46 percent of the population in these regions was covered by a SIPO at the time of Kentucky's equivalent measure. While Kentucky's policies were therefore not an outlier among the Midwest and South as a whole, it was an early mover among "white-working-class" states. Among the seven states with index values of greater than 0.75, it implemented the earliest school and restaurant/entertainment center closures and the second-earliest gathering bans and SIPOs.

## 3. Econometric model

Our analysis has three main objectives: (1) to measure the dynamic impact of social distancing restrictions across counties in the South and Midwest in the early stages of the pandemic, (2) to estimate the extent to which this effect differs in predominantly white, working-class counties in these regions, and (3) to use those results to quantify the impact of Kentucky's relatively strong and early interventions. This section describes the empirical strategy for achieving these objectives.

## Average effects

First, we estimate the average effect across all counties in the South and Midwest–as well as how that effect evolved over time–using an event-study-style generalization of the standard difference-in-differences econometric framework. Whereas difference-in-differences models include the interaction of indicators for treatment group and whether the time is post-treatment, an event-study model allows for more flexible timing of impacts by interacting treatment with *several* indicators of time relative to treatment. Differentiating between the effects of treatments that just occurred versus those that occurred a longer time ago may be particularly valuable in the case of COVID-19, where the effects of social-distancing restrictions are likely to be gradual due to incubation periods, delays seeking medical care after the onset of symptoms, delays obtaining COVID-19 tests after seeking care, and waits for test results [34]. Additionally, event-study models include indicators reflecting time *before* treatment, allowing for an evaluation of pre-treatment trends. If divergence between the trends of the treatment and control groups emerges *prior* to treatment, then the observed relationships reflect unobserved confounders or reverse causality rather than the causal effect of the intervention on the outcome.

Following our prior work [12], our event-study model contains six variables for each of the four types of social-distancing policies: whether it was implemented

- 1–5 days ago,

- 6–10 days ago,

- 11–15 days ago,

- 16 or more days ago,

- -5 to -9 days ago (i.e. will be implemented 5–9 days from now), or

- <-9 days ago (i.e. will be implemented 10 or more days from now).

Implementation in the next five days (0 to -4 days ago) is the omitted reference group to which the coefficients for the other time periods are compared. For instance, the coefficient on the variable for "SIPO was implemented 11–15 days ago" measures the effect of starting a SIPO 11–15 days ago instead of in the next five days. The Supplemental Appendix to our previous work provides formal notation for our event-study model [12].

Since we include four different types of polices together in the same model, their coefficients represent partial effects, holding all other types of policies constant. While the sequence in which the restrictions on schools, businesses, and events took effect varied across states and counties, SIPOs were almost always implemented after at least one other restriction was in place–usually all three. The estimated effects of SIPOs therefore represent their *additional* impacts above and beyond prior closures. The "full" impact of issuing a SIPO–which, by definition, encompasses the other smaller restrictions–without any prior restrictions is better measured as the linear combination of the coefficients of all four policy variables in the specified time frame.

Other covariates in our model are county and day fixed effects; i.e. dummy variables for each county and each day in the sample. The county fixed effects capture determinants of counties' COVID-19 case growth rates that do not appreciably change throughout the sample period, such as population density, demographic characteristics of the population, and political orientation [9, 35]. The day fixed effects capture common shocks to case growth rates shared by all counties in the U.S., such as voluntary social distancing in response to CDC guidance and other nationwide sources of information, international travel bans, and national trends in

testing access. Therefore, the ability to interpret our results as causal effects of the policies rests on the assumption that these unmeasured factors do not change differentially across counties over time in a way that is correlated with the timing of the policies' implementation.

In our previous work [12], we report results from numerous robustness checks designed to rule out threats to the validity of this model. Their first few checks show that their statistically insignificant results for school closures and event bans are *not* due to insufficient identifying variation independent from the other policies. Their other checks provide evidence that their main conclusions are not sensitive to excluding unique early outbreak states, constructing the policy variables in other defensible ways, imputing or excluding certain questionable observations in the case data, starting the sample at a different time, controlling for number of tests performed in the state, and controlling for county-specific pre-treatment trends in case growth rate. In unreported regressions (results available upon request), we have verified that the results are similarly robust to these checks in our dataset.

## Heterogeneous effects

The average effects across the South and Midwest likely better reflect Kentucky's experience than average effects across the entire U.S., but may not be completely applicable, as there remains heterogeneity within those regions. In particular, Kentucky is perhaps best characterized as a state that is relatively heavily "white working class", as proxied by the index of political and religious views, race/ethnicity, rurality, and education described above. Therefore, if the impact of social distancing restrictions varies with different values of the index, we can use this information to better estimate the extent to which Kentucky's decisive early action slowed the spread of COVID-19.

Our model of heterogeneous impacts simply adds the interactions of the index with each of the policy variables (both leads and lags of each of the four policies) to the event-study model. Ideally, we might like to more precisely identify the source of the heterogeneity by interacting the policies with each of the five components of the index rather than the index as a whole. However, adding interactions of each of the five county characteristics with each of the twenty-four policy variables leads to an enormous number of parameters to be estimated, while strong correlations among these characteristics make the results difficult to interpret. Since heavily Trump-leaning counties tend to be relatively rural with high percentages of non-college-educated, white, Evangelical Christians, we prefer to examine the bundle of characteristics together. Nonetheless, results from a regression that fully interacts each policy variable with each of the five characteristics are shown in the appendix.

## Implications for Kentucky

The results from the regression with interaction terms can, in principle, be used to compute predicted impacts of each policy on any county in the sample. The predicted effect of a particular policy variable (e.g. SIPO after 6–10 days) for a particular county is given by multiplying the county's index value by the coefficient on the interaction term, and adding this to the coefficient on the corresponding uninteracted policy variable. Predicted effects are particularly straightforward at index values of zero (i.e. the "average" county) and one (one standard deviation more "white working class" than average). The former is simply the coefficient on the uninteracted policy variable while the latter also adds in the coefficient on the interaction term. The latter case also corresponds closely to the state of Kentucky as a whole, as the county-population-weighted average index value in the state is 0.94.

That said, there is considerable heterogeneity even within the state of Kentucky. The two large consolidated city-counties of Louisville-Jefferson County and Lexington-Fayette County

are the only two counties in the state with below average values of the "working-class white" index, at -0.312 and -0.225, respectively. Each of the state's other 118 counties has an above average index value, ranging from 0.272 to 2.684. Simulating the impacts of social distancing restrictions separately for these two groups therefore provides a helpful illustration of the potential for heterogeneous effects even within relatively "white working-class" states.

It is important to note that, while Kentucky is our focal state for the purpose of illustration, the identifying variation in our policy variables and interaction terms comes from the entire South and Midwest. This is in contrast to, for instance, a synthetic control analysis in which Kentucky is the treated state and a composite of other similar states serves as the control group. There are two main reasons for this. First, every state in the U.S. enacted multiple social distancing restrictions–and in most cases all four of the ones we consider. The identifying variation therefore comes from the timing of treatment; there is no true control group. Comparative case study methods such as synthetic control are inherently poorly suited for situations where multiple treatments were enacted at multiple points in time in multiple places. Second, utilizing identifying variation from multiple regions containing dozens of states gives the results more generalizability beyond Kentucky. While practical considerations prevent us from showing the effects of several policies at several points in time in the more than 2,000 counties in our sample, state or local policymakers interested in using our results to simulate impacts in their jurisdiction could easily do so.

## 4. Results

### Average effects

The coefficients and their 95% confidence intervals for the four social-distancing policies obtained from the event-study are displayed in Fig 2. The confidence intervals are based on standard errors that are robust to heteroskedasticity and clustering by state. A variable is statistically significant at the 5% level if its coefficient's confidence interval does not include zero.

The upper left panel of Fig 2 shows that SIPOs led to a gradual but substantial reduction in the growth rate of COVID-19 cases. In the first time period after the implementation (1–5 days), the growth rate fell by a statistically insignificant 2.4 percentage points relative to the reference period of the five days before implementation. The effect became statistically significant in each subsequent period, growing to 3.9 percentage points after 6–10 days, 5.3 percentage points after 11–15 days, and 8 percentage points after 16 or more days.

The restrictions on restaurants and entertainment centers had a relatively steady effect on the growth rate of COVID-19 cases once in place. Closing either restaurant dining areas or gyms/entertainment centers (usually both) lowered the growth rate by 5 percentage points in the first five days after taking effect, with the impact rising only slightly–to 6.2 percentage points–after 16 days. In contrast, however, we found no evidence that bans on large social gatherings or school closures reduced growth rates after any length of time, holding the other types of policies constant. The estimated effect of event bans is nearly zero and statistically insignificant, while the effect of closing schools is consistently positive (faster case growth), though never significant.

For all four types of policies, we observe no "placebo" effect on the pre-enactment growth rate. All coefficient estimates representing "impacts" of future implementation are small and statistically insignificant. In other words, we find no evidence of bias from reverse causality (case growth driving policy implementation) or unobserved confounders.

As discussed previously, the combined effect of all four policies is the best estimate of the overall impact of government-imposed social distancing restrictions. This combined effect was 5.4 percentage points in the first five days, 6.8 percentage points after 6–10 days, 7.1 percentage

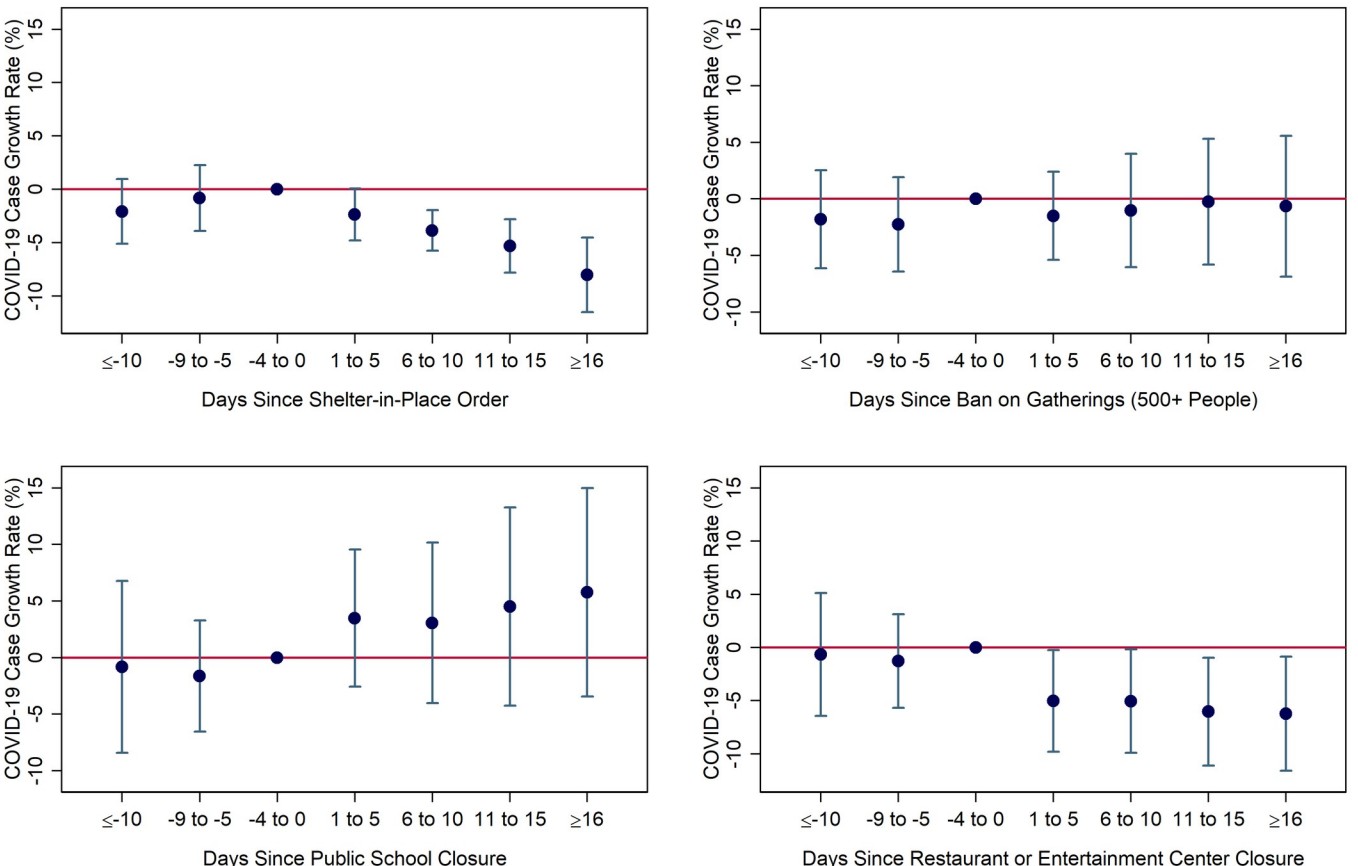

**Fig 2. Estimated impacts of social-distancing measures over time from event-study model.** Notes: Coefficients and 95% confidence intervals were derived from authors' event-study regression using daily county-level data from March 5-April 25 from the South and Midwest Census Regions, weighted by population. Day and county fixed effects were included. Standard errors were heteroskedasticity-robust and clustered by state.

points after 11–15 days, and 9.1 percentage points after 16 or more days. Only the second and fourth of these estimates is statistically significant at the 5% level, with the first and third having p-values of 0.16 and 0.12, respectively. These estimates are relatively similar to those for SIPOs alone, as the positive (though insignificant) effect of closing schools on case growth was roughly offset by the reduction from restaurant/gym/entertainment center closures. In other words, on average, states' and counties' early social distancing efforts made little difference in slowing the spread of COVID-19, but stronger measures implemented later were more successful.

## Heterogeneous effects

We next present the results from our interaction model. In Fig 3, we report the coefficient estimates for the non-interacted social-distancing policy indicators, which can be interpreted as the effects in counties with average levels (and therefore values of zero) of the "white working-class" index. Broadly, the results echo those from Fig 2, as SIPOs and closures of restaurants and entertainment-related businesses lead to large and statistically significant reductions in

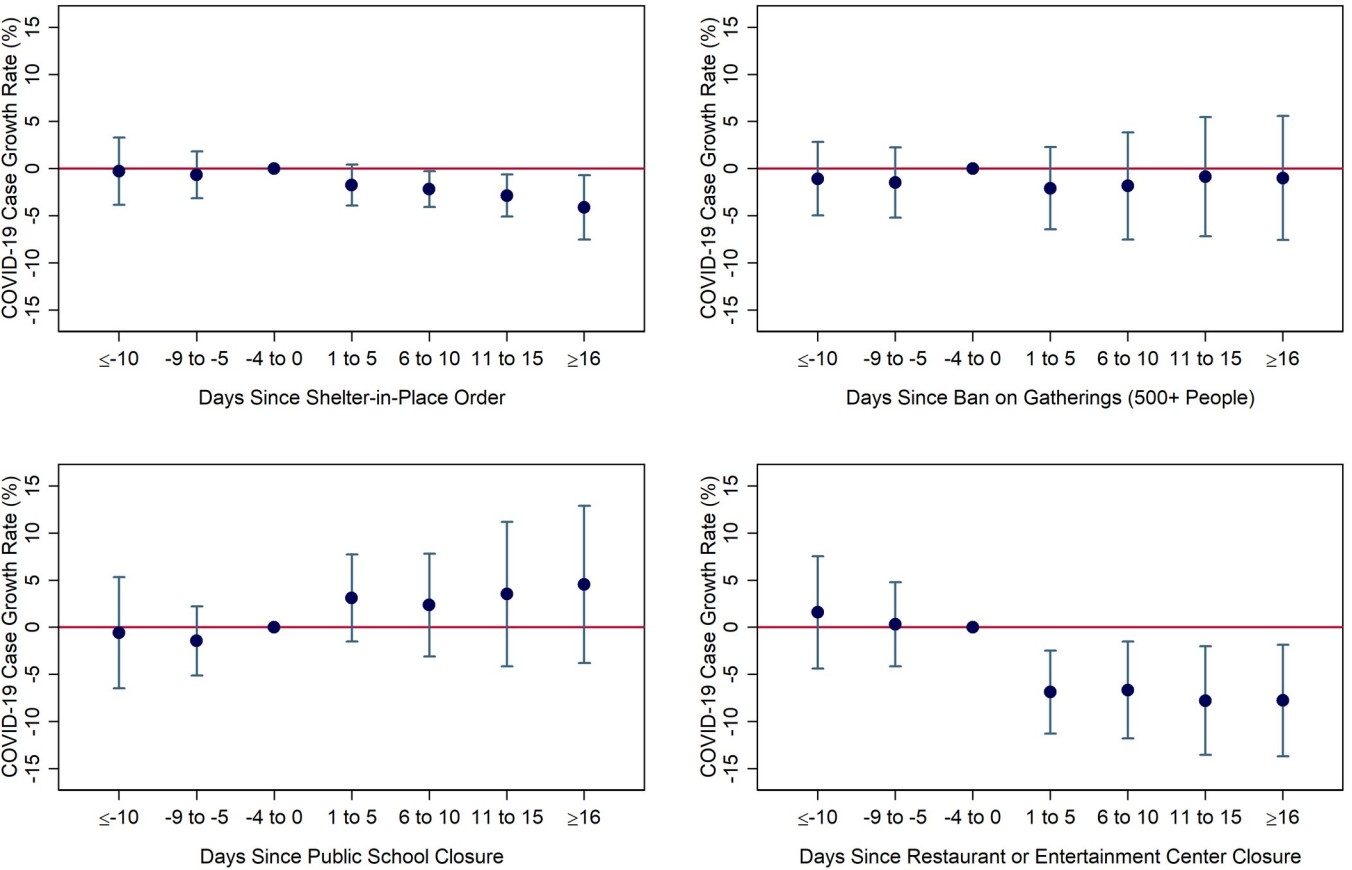

**Fig 3. Coefficient estimates for uninteracted social-distancing measures over time from model including index.** Notes: Coefficients and 95% confidence intervals were derived from authors' event-study regression using daily county-level data from March 5-April 25 from the South and Midwest Census Regions, weighted by population. Day and county fixed effects were included. Standard errors were heteroskedasticity-robust and clustered by state. Specifications include interactions between regulation indicators and the "white working class" index variable.

the COVID-19 daily case growth rate, while school closures and large gathering bans had no statistically detectable effects.

Fig 4 shows the estimates for the interaction terms, which represent the additional effects (on top of those shown in Fig 3) in places that are one-standard-deviation more "white working class" than average, such as the state of Kentucky as a whole. The estimates are largely a mirror image of the uninteracted effects. The interactions of the index with both SIPOs and entertainment-related closures are positive and statistically significant in all periods, with magnitudes that are similar—but the opposite sign—of those observed for the uninteracted variables. The interactions for school closures and gathering bans are statistically insignificant in the post-treatment periods, as was the case for their uninteracted counterparts.

On net, then, these results imply that social distancing restrictions had little discernable impact in counties with one-standard-deviation-higher-than-average "white working-class" characteristics. Specifically, the interaction effect more than negated the entire (uninteracted) effect of SIPOs in the first five days after implementation, negated 70 percent of the effect from days 6 to 10, negated 78 percent in days 11 to 15, and negated 63 percent in days 16 and

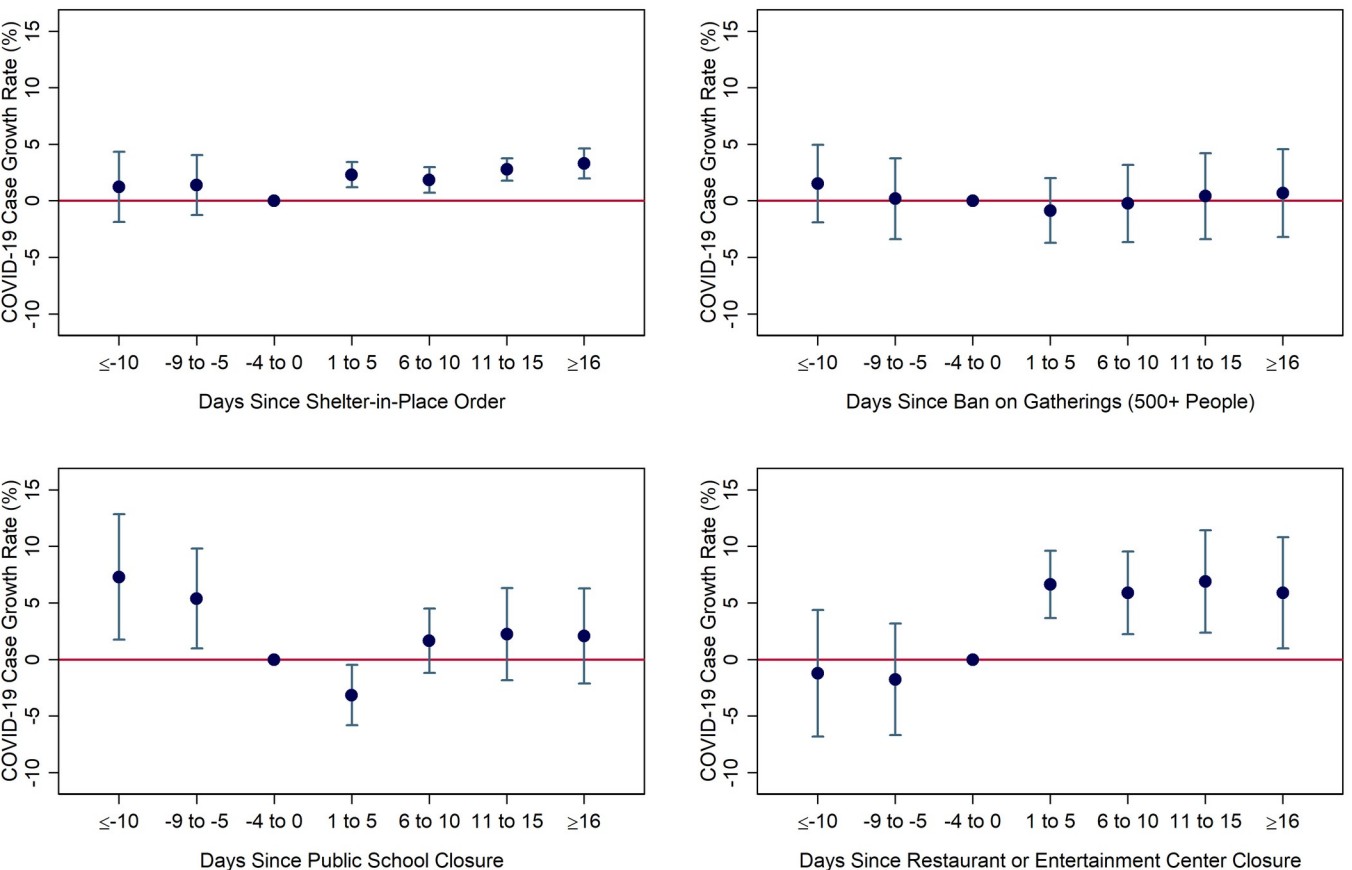

**Fig 4. Coefficient estimates for interactions of the "white working class" index with social-distancing measures.** Notes: Coefficients and 95% confidence intervals were derived from authors' event-study regression using daily county-level data from March 5-April 25 from the South and Midwest Census Regions, weighted by population. Day and county fixed effects were included. Standard errors were heteroskedasticity-robust and clustered by state. Specifications include interactions between regulations indicators and the "white working-class" index variable.

onward. The remaining effect after 16+ days was less than a percentage point and statistically insignificant. For restaurant/entertainment closures, the interaction effect negated most or all of the impact in the first 15 days and 75% of the effect after 16 days, leaving a remaining effect of fewer than two percentage points.

## Implications for Kentucky

We next illustrate these patterns more clearly by showing the net effect (from both the uninteracted and interacted policy variables) of social distancing restrictions in Louisville/Lexington and the rest of Kentucky, respectively. In Fig 5, we see that the regression estimates imply that SIPOs and restaurant/entertainment center closures helped to flatten the COVID-19 curve in Louisville and Lexington. SIPOs' impact gradually strengthened over time, reaching 5 percentage points (p<0.01) after 16 days. Restaurant/entertainment center closures reduced the daily growth rate by 8–10 percentage points in all post-treatment time periods. The combined effect of all four policies reached 11.75 percentage points after 16 days–a larger reduction that that observed in the baseline analysis without interaction terms. This is because the average index

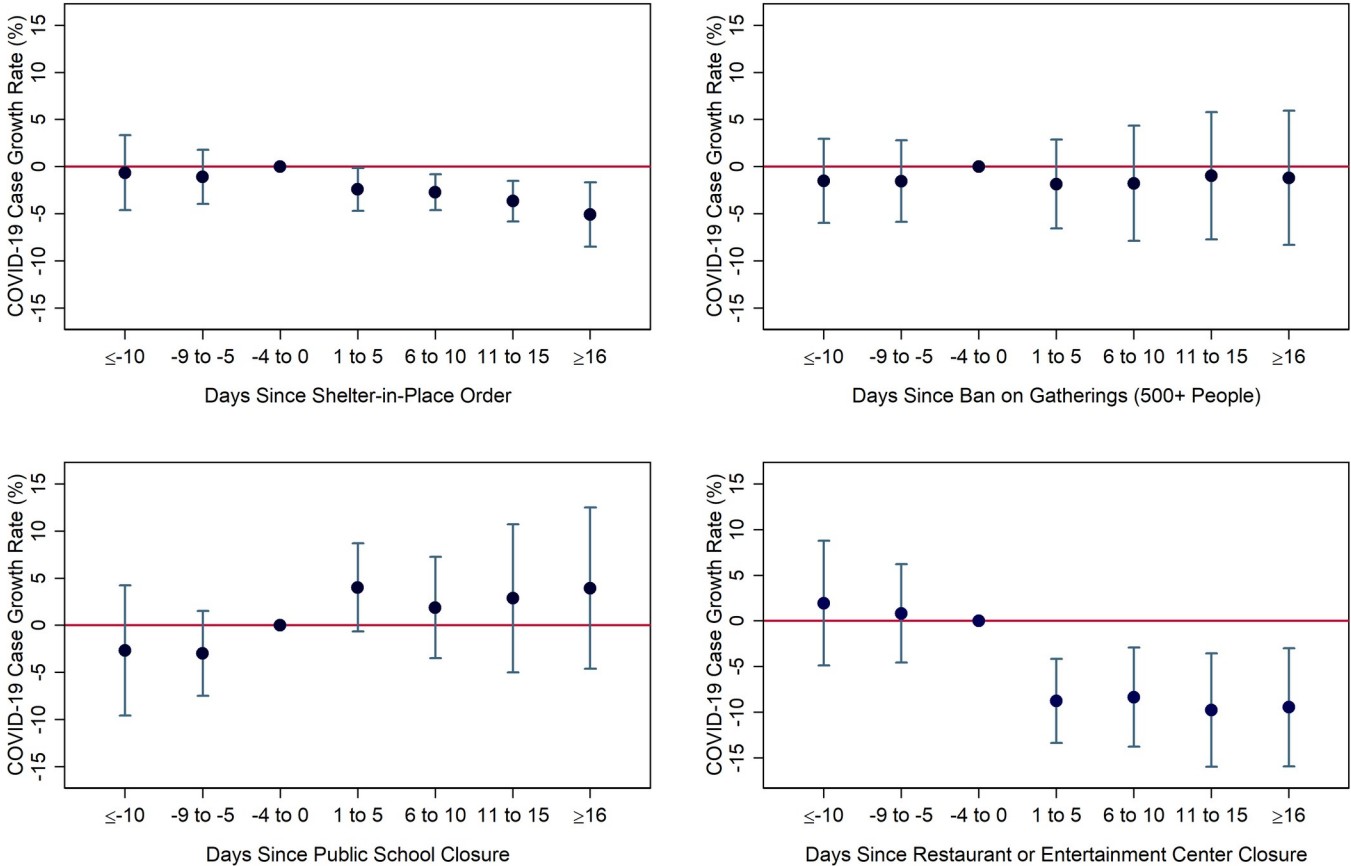

**Fig 5. Predicted effects of social-distancing measures in Louisville-Jefferson County and Lexington-Fayette County.** Notes: Coefficients and 95% confidence intervals were derived from authors' event-study regression using daily county-level data from March 5-April 25 from the South and Midwest Census Regions, weighted by population. Day and county fixed effects were included. Standard errors were heteroskedasticity-robust and clustered by state. Specifications include interactions between the regulation indicators and the "white working-class" index variable. Results shown are linear combinations of uninteracted and interacted policy variables, evaluated at the average index value of -0.29 in the two cities.

value across Louisville and Lexington is -0.29, so the coefficients on the interaction terms imply that these cities should be more strongly affected than the average county.

In contrast, the rest of Kentucky has an average index value of 1.34, implying that the projected impacts are even weaker than those computed above at an index of one. Indeed, Fig 6 shows that none of the policy variables are ever statistically significant at the 5% level in any post-treatment period. The combined effect after 16+ days is actually positive, though insignificant, as it is driven by the large but insignificant positive coefficient estimate for school closures. In short, there is no evidence that social distancing restrictions flattened the COVID-19 curve in Kentucky at all, outside of the two largest, most liberal, and most ethically and religiously diverse urban counties.

## 5. Conclusion

Our results show that strong social distancing policies have, on average, strong effects in Southern and Midwestern states. However, an important source of heterogeneity emerges

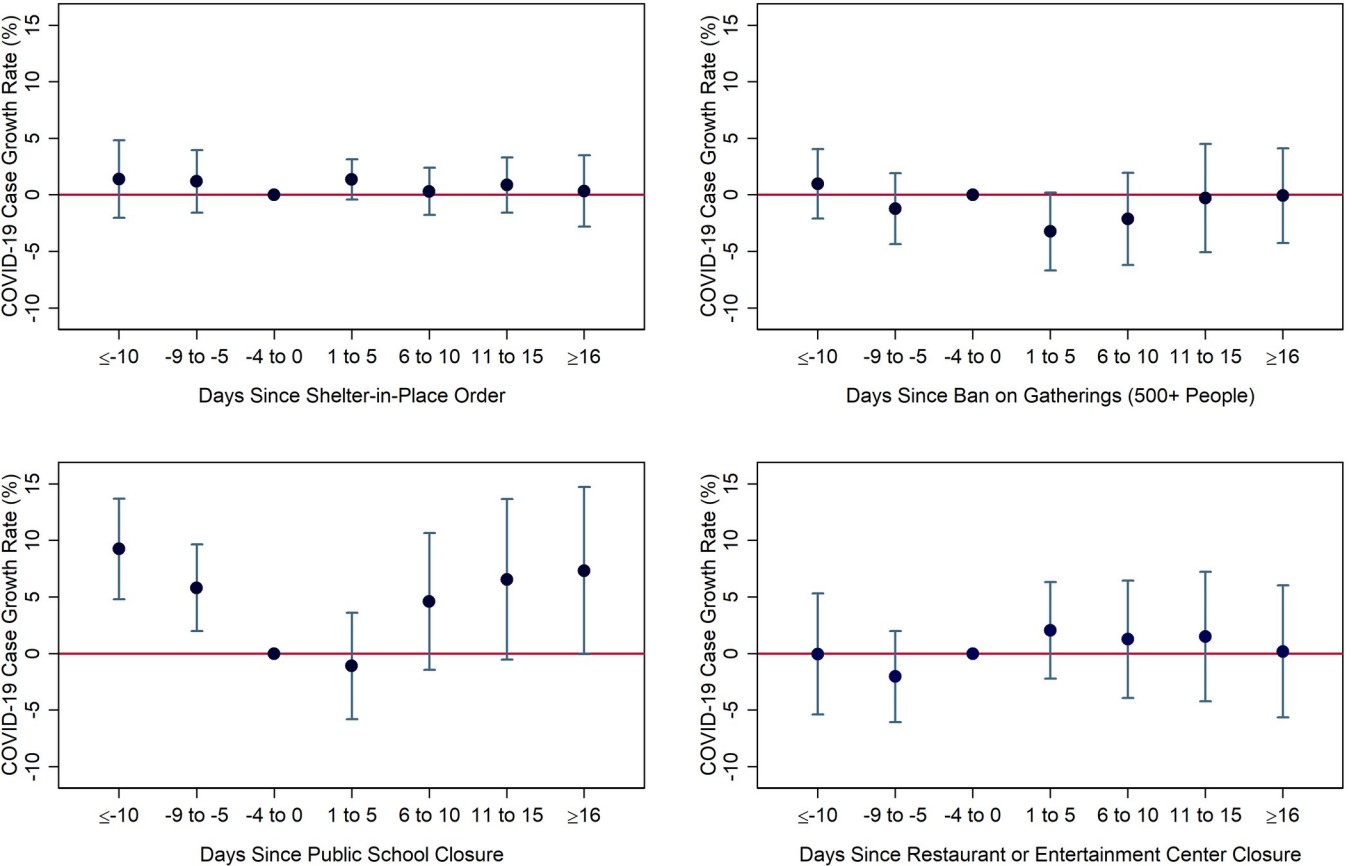

**Fig 6. Predicted effects of social-distancing measures in Kentucky outside of Louisville and Lexington.** Notes: Coefficients and 95% confidence intervals were derived from authors' event-study regression using daily county-level data from March 5-April 25 from the South and Midwest Census Regions, weighted by population. Day and county fixed effects were included. Standard errors were heteroskedasticity-robust and clustered by state. Specifications include interactions between regulations indicators and the "white working-class" index variable. Results shown are linear combinations of uninteracted and interacted policy variables, evaluated at the average index value of 1.34 in Kentucky (excluding Lexington and Louisville).

where the strong social distancing measures are most effective in more densely populated areas with lower shares of white, politically conservative, Evangelical Christian residents and relatively low rates of college completion. We examine the implications of these results for Kentucky–a predominantly white, working-class state where a coin-flip election resulted in a Democratic governor implementing earlier restrictions than other similar states. Our findings imply that Kentucky's aggressive early action was likely only effective in flattening the COVID-19 curve in the state's largest cities of Louisville and Lexington.

Our finding of strong effects on average aligns closely with other recent, quasi-experimental studies on the impact of social distancing measures in California [11] and across the U.S. as a whole [12, 13]. Our results are also consistent with evidence that SIPOs and closures of restaurants/entertainment facilities reduce cell-phone-tracked measures of mobility, while school closures and bans on medium-sized events do not [6, 7].

On the other hand, one might wonder how our results based on COVID-19 cases reconcile with some studies [8, 36], that suggest reductions in mobility largely occurred prior to SIPOs. First, measures of human mobility from cell-phone data are imperfect proxies of social

distancing behaviors. The most common measure used from SafeGraph data–whether an individual stays within a "home" area for the entire day–does not capture the precautions one may take when interacting with other individuals, and becomes a worse proxy for potential interactions as population density increases. "Home" is an area of roughly 153m x 153m. If population density can affect the reproduction rate of a respiratory virus, we should be wary of a proxy for social distancing that allows considerable room for interactions in denser urban areas and no room for interactions in many rural areas. Second, as emphasized in other work [11], small government-induced changes in mobility can yield large changes in public health, especially when the disease is growing exponentially. Finally, the marginal compliers who voluntarily respond to initial information events about the dangers of the virus may be different than those who comply with government mandates. In practice, one likely scenario is that the reproduction rate varies with an individual's characteristics, and those whose compliance was induced by mandates may have had more social interactions to begin with.

Our finding of relatively weak effects in "white working-class" areas is consistent with current work [13] finding that SIPOs were most effective in early adopting states and those with high population densities, and other work [14] concluding that SIPOs in Texas were more effective in urbanized counties than other parts of the state. It is perhaps not surprising that compliance with policies like SIPOs that are effectively unenforceable social nudges would be weaker in areas where the residents are generally less supportive of social distancing restrictions. However, our analysis moves beyond SIPOs and also documents heterogeneous effects of restaurant and entertainment facility closures, which are more readily enforceable. This suggests an additional mechanism through which weaker effects arise in "white working-class" localities. Perhaps such places simply have fewer restaurants and entertainment establishments than more affluent urban areas, so daily life is not as strongly affected when they are forced to close. An explanation like this points to the ongoing difficulty with controlling the pandemic when most of the social interactions that occur in a community are in private settings where public health guidelines cannot be enforced.

Most SIPOs in the states we examined expired within a few months of their start dates. Georgia, which instituted a SIPO on April 3, allowed gyms, hair and nail salons, bowling alleys and tattoo parlors to reopen on April 24; and reopened restaurants, movie theaters, and other entertainment on April 27. Tennessee's SIPO expired on April 30, but restaurants started reopening at reduced capacity on April 27. Consistent with the aggressive initial response, Kentucky started reopening later, on May 11, and more cautiously, with restrictions on restaurants not relaxing until May 22. However, all of these states eased restrictions earlier than many public health experts advised, and the well-documented explosion of cases in the subsequent months is consistent with our finding that at least some types of restrictions had strong causal effects in at least some areas. The Institute for Health Metrics and Evaluation projected June 14 as the day Kentucky could consider relaxing social distancing measures with a containment strategy [5]. Some of the states that relaxed restrictions earlier than Kentucky–Georgia, Oklahoma, South Carolina, and Tennessee–had projections of June 22, June 17, June 8, and May 20, respectively. According to those projections, few states were positioned to reopen when they did, even with a containment strategy [37].

State policymakers have also proven more willing to reopen restaurants than to reopen schools. We find no evidence that school closures and bans of moderately-sized events have *any effect at all* on the growth rate of COVID-19 cases unless accompanied by broader shelter-in-place directives such as Kentucky's "Healthy at Home" initiative. On the other hand, we did find that closing bars, restaurants and gyms prior to instituting SIPOs played some role in slowing the spread of the virus.

Thinking about why closing schools and banning large gatherings had no statistically detectable effects is also informative. First of all, our finding on school closures is very much consistent with contact tracing studies that find transmission from children to adults is relatively rare [38]. Meanwhile, the null result for large gathering bans could simply mean that potential super-spreader events (e.g. college and professional basketball games) were already voluntarily being cancelled even before the official restrictions. An alternative, behavioral explanation for our findings on both school closures and bans on events comes from the economic concept of substitution. Parents who cannot send children to school may take their kids to visit friends, out to eat lunch, or to run errands. People who can't attend a concert or sporting event may go to a bar instead. Recent studies show that potential super-spreader events–indoor political rallies by former President Trump and widespread Black Lives Matter protests–led to a number of responses (including voluntary restaurant and bar closures, and curtailing of typical activities) that in turn moderated the effect on community spread [39, 40]. Ultimately, slowing the spread of COVID-19 depends on the degree to which policies *reduce* social interactions, not on whether they displace social interactions. All that said, it is also important to remember that lack of evidence of an effect is not the same as conclusive evidence of no effect. All but one state closed public schools between March 16 and March 23 –a narrower range than the other policy variables. Accordingly, the confidence intervals associated with the school closure coefficient estimates are relatively large and generally cannot rule out moderately large effects. It is therefore not clear how informative these results are for ongoing school reopening decisions.

Finally, a natural question to ask is how our findings from early in the pandemic relate the situation in early 2021, when we are in the early stages of a vaccine rollout and face increased fatigue with lockdown measures. While weariness with lockdowns seems more widespread, the population was more antagonistic to lockdown measures early in the pandemic in the South and Midwest than in other parts of the country. In late 2020, California imposed a stay-at-home-order but abruptly lifted it in early 2021, despite surging cases in parts of the state. That reversal almost certainly reflected political considerations rather than public health concerns. Recent work finds that, even in the current environment with a vaccine rollout, an economic lockdown of 2 to 4 weeks would generate net benefit of up to $1.2 trillion, mostly by avoiding fatalities [41]. Our results here, based on a sample of relatively conservative states, suggests that strong social distancing policies can still be effective in bending the curve. However, such effectiveness should not be taken for granted, and likely depends on both attitudes in the community about the restrictions' merit, as well as underlying characteristics of the area and its residents.

## Author Contributions

**Conceptualization:** Charles Courtemanche, Aaron Yelowitz.

**Data curation:** Charles Courtemanche, Joseph Garuccio, Anh Le, Aaron Yelowitz.

**Formal analysis:** Joseph Garuccio, Anh Le, Joshua Pinkston.

**Investigation:** Charles Courtemanche.

**Methodology:** Charles Courtemanche, Joshua Pinkston.

**Project administration:** Charles Courtemanche, Aaron Yelowitz.

**Supervision:** Charles Courtemanche, Aaron Yelowitz.

**Validation:** Charles Courtemanche, Joseph Garuccio, Anh Le, Joshua Pinkston.

**Visualization:** Joseph Garuccio, Anh Le.

**Writing – original draft:** Charles Courtemanche, Joseph Garuccio, Anh Le, Joshua Pinkston, Aaron Yelowitz.

**Writing – review & editing:** Charles Courtemanche, Joseph Garuccio, Anh Le, Joshua Pinkston, Aaron Yelowitz.

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
