## [Decision Letter · Decision Letter 0]

17 Aug 2020

PONE-D-20-12500

Did Social-Distancing Measures in Kentucky Help to Flatten the COVID-19 Curve?

PLOS ONE

Dear Dr. Courtemanche,

Thank you for submitting your manuscript to PLOS ONE. After careful consideration, we feel that it has merit but does not fully meet PLOS ONE’s publication criteria as it currently stands. Therefore, we invite you to submit a revised version of the manuscript that addresses the points raised during the review process.

Both reviewers agree that the paper is well written and the analysis well-executed, but both reviewers point out that the contribution of this manuscript is minor given your previously published work. After reading the manuscript, I agree that this manuscript is too similar to your previous work to warrant a separate publication. However, the second reviewer makes some excellent suggestions on how to differentiate this manuscript from your previous work and I would like to see a revised manuscript that do that.  

We look forward to receiving your revised manuscript.

Kind regards,

Gabriel A. Picone

Academic Editor

PLOS ONE

Journal Requirements:

2. Please note that PLOS ONE does not allow for footnotes in its publications. As such, we ask you to remove these from your manuscript and move this information to the main text.

Reviewers' comments:

Reviewer's Responses to Questions

**Comments to the Author**

1. Is the manuscript technically sound, and do the data support the conclusions?

Reviewer #1: Yes

Reviewer #2: Partly

2. Has the statistical analysis been performed appropriately and rigorously? 

Reviewer #1: Yes

Reviewer #2: Yes

3. Have the authors made all data underlying the findings in their manuscript fully available?

Reviewer #1: Yes

Reviewer #2: Yes

4. Is the manuscript presented in an intelligible fashion and written in standard English?

Reviewer #1: Yes

Reviewer #2: Yes

5. Review Comments to the Author

Reviewer #1: This is a well written piece and has clearly benefitted from your work on the earlier paper.

I am not completely sure of the contribution over your earlier paper, other than extended time period and a more narrow focus. You may want to be more clear about what is different.

I think one thing in your conclusion that was not in the earlier paper is the discussion about the correlation of restrictions. This is actually the most important discussion I believe as papers, including this one, are not finding school closures are helpful. Is that because the school closures all occurred simultaneously with the other restrictions or was it because school closures lagged changing behavior of locals? I am not sure you need to do that is this short paper but it would be an interesting idea to explore with your data. This is clearly an important topic and I am concerned that what you have found can be taken out of context to support school openings when I am not certain that is your finding. Maybe address this issues a bit more earlier on in the paper - which you present the event studies.

I enjoyed reading your paper.

Reviewer #2: Report: “Did Social-Distancing Measures in Kentucky Help to

Flatten the COVID-19 Curve?”

Overview

Emerging literatures in economics and public health find that state and local shelter-in-place orders (SIPOs) were effective in curbing COVID-19 case growth, particularly if enacted early in the outbreak cycle. This study contributes to these literatures with a case study of Kentucky. Specifically, the study explores the comparative effectiveness of a series of policies adopted in Kentucky to fight COVID-19 spread: public school closures, restaurant dining room closures, entertainment center closures, non-essential business closures, and a “Healthy at Home” (HAH) order. Event-study analyses show that the HAH order was most effective at curbing COVID-19, followed by restaurant/ entertainment center closure. The authors estimate that over 40,000 COVID-19 cases were averted in Kentucky due to social distancing policies adopted.

Comments

1. Given that there is substantial heterogeneity in both (i) state and local social distancing policies, and (ii) responses to such policies (due to differences in local populations), I believe that there is strong value-added in case studies of particular state (or local) policies. Therefore, it does not bother me that other published studies (including one by this team) have pooled Kentucky with other states/counties and estimated COVID-19 case growth effects of social distancing policies.

However, the authors could do a better job of making the case for “Why Kentucky?” Is there something unique about the policies (i.e., the substance of the orders, the timing of the orders, the time between each order, the legal enforceability of the orders?) in Kentucky compared to other states that makes them especially interesting? Is there something unique about the population that could generate unique policy responses? Making this case forcefully in the Introduction would help convince readers of the paper’s contribution relative to the existing literature.

2. The research design of this study surprised me a little. Given the title, I was expecting the authors to “directly” estimate the effect of Kentucky’s social distancing policies on COVID-19 case growth in Kentucky. Instead, the authors pool Kentucky with other states (as part of two census regions), estimate average treatment effects (across all jurisdictions in these regions) in an event-study framework, and then use these estimates to infer Kentucky’s COVID-19 case avoidance. (This approach is very similar to the approach taken by Courtemanche et al. 2020 in their Health Affairs paper – the main difference is that here, the authors obtain estimates of social distancing policy impacts using a restricted sample of the Midwest and South Census Regions.)

My question is: why not directly estimate the effect of Kentucky’s policies? For example, why not estimate the effects of KY’s HAH order using, for example, a synthetic control design? The other social distancing policies could be accounted for by “matching” on the share of days over the sample period that the policies were in effect.

Identifying donor states should be straightforward for the HAH policy (non-SIPO adopters and states that adopted SIPOs at least X days (5? 7?) after Kentucky’s order. I concede that finding credible donors for some of the other policies may be difficult. For example, donors for the “public school closure” policy evaluation would have to rely on states that closed schools late given that these closures were nearly universal. Local policies create another complication. But a synthetic control approach seems like the most obvious research design to employ first. Relatedly, the authors could estimate a difference-in-difference (or event study) model using a sample consisting of Kentucky (the one treatment state) and the donor states. If synthetic controls are not an option to study Kentucky’s policies – perhaps because the donor pool is too limited and not credible — this could be explained in a lengthy footnote so readers know this was considered and rejected.

A second advantage of pursuing an alternative estimation strategy (if it worked out!) is that it would make the value-added of the study larger. It would avoid the problem of this paper being seen as a “small tinkering” of the Health Affairs paper.

3. Is the outcome in this paper the same as the outcome in the Health Affairs paper? In the HA paper, the authors write:

“The daily exponential growth rate was calculated as the natural log of cumulative daily COVID-19 cases minus the log of cumulative daily COVID-19 cases on the prior day.” (p. 1238)

In this paper, the outcome is described as:

“We use these data to compute each county’s daily exponential growth rate in confirmed COVID-19 cases, which is equal to the natural log of daily COVID-19 cases minus the log of daily COVID-19 cases on the prior day.” (p. 6)

Why the change from growth in cumulative daily cases to growth in daily cases? A case can be made for looking at either margin (growth in day to day cumulative cases vs day to day daily cases)…both are interesting. Readers should just understand the comparability or non-comparability of the estimates across these two papers. Or maybe it was just a typo?

4. The authors might mention information as an important channel through which some of these policies could affect social distancing and COVID-19 case growth. For instance, a HAH order may send important information to residents about the seriousness of the epidemic (or scare the heck out of people), which could increase stay-at-home behavior. Staying-at-home may also facilitate information gathering by watching more news reports (i.e. Daily White House Coronavirus Briefings, etc.)

5. Finally, the authors claim that in the absence of social distancing policies, COVID-19 cases in Kentucky would be an order of magnitude (11 to 12 times) higher. This a is a very large number and one that is likely to draw the attention of those skeptical of large COVID case effects of SIPOs. As the authors know, there is a recent set of studies (Cronin and Evans 2020; Goolsbee and Syverson 2020; Gupta et al. 2020; Sears et al. 2020) that emphasize that most of the variation in social distancing behavior comes from “private responses” to information/risk/beliefs rather than responses to SIPOs. (Still, plenty of credible evidence shows that SIPOs have an effect over and above these private responses.) However, given this set of papers, the authors might want to frame their effect sizes in terms of what we know about (i) social distancing elasticities with respect to the policies they study, and (ii) plausible COVID-19 case elasticities with respect to social distancing. This comment is NOT meant as an instruction to reconcile this paper’s findings with the above set of papers. Rather, the comment is designed to help the authors frame their magnitudes in the context of this growing literature.

6. PLOS authors have the option to publish the peer review history of their article (what does this mean?). If published, this will include your full peer review and any attached files.

Reviewer #1: No

Reviewer #2: No

---

## [Author Response · Author response to Decision Letter 0]

16 Feb 2021

Response to Reviewer #1

Thank you for taking the time to review our paper. We appreciate your comments and feel that addressing them in your revision has improved the paper substantially. Your comments are printed verbatim below, followed by our responses.

This is a well written piece and has clearly benefitted from your work on the earlier paper.

Response: Thank you! 

I am not completely sure of the contribution over your earlier paper, other than extended time period and a more narrow focus. You may want to be more clear about what is different.

Response: You are correct – the only meaningful difference in the analyses between our Health Affairs paper and the initial version of this paper was that the sample was restricted to the Midwest and South, with the motivation being to obtain results that were relevant to relatively rural, politically conservative states like Kentucky.

In the revised version of the paper, we have sharpened this contribution in a way that should make the novelty much more clear:

(1) We have rewritten much of the introduction to better articulate why it is important to obtain results specific to right-leaning, relatively rural, predominantly “white working class” states like Kentucky. In other words, why can we not just assume results obtained using a nationwide sample are applicable? In short, (i) shelter-in-place orders (SIPOs) are effectively unenforceable nudges that rely on social pressure, so underlying attitudes about their appropriateness likely influence compliance; (ii) restaurant and entertainment establishment closures might have less of an impact on small town “working class” communities simply because such establishments are less prevalent there. 

(2) We have added a new empirical analysis to capture this heterogeneity more directly. We developed a county-level “white working-class” index that is based on the county’s population density and percentages of residents who are white, Evangelical Christian, voted for President Trump, and do not have college degrees. We then re-estimate the regression model adding interactions of this index with each of the policy variables. The results show that the effectiveness of social distancing restrictions decreases markedly with higher values of the index. The results imply that the restrictions did little to flatten the curve in counties that are one standard deviation more “white working class” than average. We feel that this result adds important nuance to a literature that currently, on balance, argues that SIPOs and restaurant/entertainment facility closures were quite effective.

I think one thing in your conclusion that was not in the earlier paper is the discussion about the correlation of restrictions. This is actually the most important discussion I believe as papers, including this one, are not finding school closures are helpful. Is that because the school closures all occurred simultaneously with the other restrictions or was it because school closures lagged changing behavior of locals? I am not sure you need to do that is this short paper but it would be an interesting idea to explore with your data. This is clearly an important topic and I am concerned that what you have found can be taken out of context to support school openings when I am not certain that is your finding. Maybe address this issues a bit more earlier on in the paper - which you present the event studies.

Response: This is a good point, and we have added more discussion of the implications and caveats of the school closure result in the conclusion on pages 23 and 24. In particular, we now have a cautionary note that states, 

… it is also important to remember that lack of evidence of an effect is not the same as conclusive evidence of no effect. All but one state closed public schools between March 16 and March 23 – a narrower range than the other policy variables. Accordingly, the confidence intervals associated with the school closure coefficient estimates are relatively large and generally cannot rule out moderately large effects. It is therefore not clear how informative these results are for ongoing school reopening decisions. 

I enjoyed reading your paper.

Response: Thank you! We appreciate the feedback.

 

Response to Reviewer #2

Thank you for taking the time to review our paper. We appreciate your comments and feel that addressing them in your revision has improved the paper substantially. Your comments are printed verbatim below, followed by our responses.

Overview

Emerging literatures in economics and public health find that state and local shelter-in-place orders (SIPOs) were effective in curbing COVID-19 case growth, particularly if enacted early in the outbreak cycle. This study contributes to these literatures with a case study of Kentucky. Specifically, the study explores the comparative effectiveness of a series of policies adopted in Kentucky to fight COVID-19 spread: public school closures, restaurant dining room closures, entertainment center closures, non-essential business closures, and a “Healthy at Home” (HAH) order. Event-study analyses show that the HAH order was most effective at curbing COVID-19, followed by restaurant/ entertainment center closure. The authors estimate that over 40,000 COVID-19 cases were averted in Kentucky due to social distancing policies adopted.

Comments

1. Given that there is substantial heterogeneity in both (i) state and local social distancing policies, and (ii) responses to such policies (due to differences in local populations), I believe that there is strong value-added in case studies of particular state (or local) policies. Therefore, it does not bother me that other published studies (including one by this team) have pooled Kentucky with other states/counties and estimated COVID-19 case growth effects of social distancing policies.

However, the authors could do a better job of making the case for “Why Kentucky?” Is there something unique about the policies (i.e., the substance of the orders, the timing of the orders, the time between each order, the legal enforceability of the orders?) in Kentucky compared to other states that makes them especially interesting? Is there something unique about the population that could generate unique policy responses? Making this case forcefully in the Introduction would help convince readers of the paper’s contribution relative to the existing literature.

Response: You are correct that the initial version of the manuscript’s contribution to the literature was relatively modest, as our empirical analysis only meaningfully differed from that of our Health Affairs paper by restricting the sample to the Midwest and South, with the motivation being to obtain results that were relevant to relatively rural, politically conservative states like Kentucky. 

In the revised version of the paper, we have sharpened this contribution in a way that should make the novelty much more clear:

(1) We have rewritten much of the introduction to better articulate why it is important to obtain results specific to right-leaning, relatively rural, predominantly “white working class” states like Kentucky. In other words, why can we not just assume results obtained using a nationwide sample are applicable? In short, (i) shelter-in-place orders (SIPOs) are effectively unenforceable nudges that rely on social pressure, so underlying attitudes about their appropriateness likely influence compliance; (ii) restaurant and entertainment establishment closures might have less of an impact on small town “working class” communities simply because such establishments are less prevalent there. 

(2) We have added a new empirical analysis to capture this heterogeneity more directly. We developed a county-level “white working-class” index that is based on the county’s population density and percentages of residents who are white, Evangelical Christian, voted for President Trump, and do not have college degrees. We then re-estimate the regression model adding interactions of this index with each of the policy variables. The results show that the effectiveness of social distancing restrictions decreases markedly with higher values of the index. The results imply that the restrictions did little to flatten the curve in counties that are one standard deviation more “white working class” than average. We feel that this result adds important nuance to a literature that currently, on balance, argues that SIPOs and restaurant/entertainment facility closures were quite effective.

We have also added new exposition and analyses to better make the case that Kentucky’s early COVID-19 experience stands out even among Midwestern and Southern states as interesting and important:

(1) The abstract and introduction now explain how a unique set of circumstances made Kentucky “as good as randomly” assigned to relatively strong, early restrictions. The confluence of a uniquely unpopular Republican incumbent governor and the Democratic challenger being the son of a former governor led to the Democrat winning the 2019 gubernatorial race in a “coin-flip” election in an otherwise deeply red state. 

(2) The abstract and introduction now also explain that Kentucky is heavily “white working class” even after restricting the sample to the Midwest and South (its index value is roughly one standard deviation above the average for those regions). Therefore, the arguments about heterogeneity made above – and corresponding regression that adds interactions of the index with the policy variables – are particularly relevant to Kentucky.

(3) We have added simulations that show how the interaction effects imply that Kentucky’s experience with COVID-19 restrictions was much different in the two biggest cities of Louisville and Lexington than in the rest of the state. Specifically, we show that the restrictions flattened the COVID-19 curves substantially in Louisville and Lexington, given their below-average “white working-class” index values. However, the restrictions had little to no effect in the other counties of the state, which all have above-average index values. 

2. The research design of this study surprised me a little. Given the title, I was expecting the authors to “directly” estimate the effect of Kentucky’s social distancing policies on COVID-19 case growth in Kentucky. Instead, the authors pool Kentucky with other states (as part of two census regions), estimate average treatment effects (across all jurisdictions in these regions) in an event-study framework, and then use these estimates to infer Kentucky’s COVID-19 case avoidance. (This approach is very similar to the approach taken by Courtemanche et al. 2020 in their Health Affairs paper – the main difference is that here, the authors obtain estimates of social distancing policy impacts using a restricted sample of the Midwest and South Census Regions.)

My question is: why not directly estimate the effect of Kentucky’s policies? For example, why not estimate the effects of KY’s HAH order using, for example, a synthetic control design? The other social distancing policies could be accounted for by “matching” on the share of days over the sample period that the policies were in effect.

Identifying donor states should be straightforward for the HAH policy (non-SIPO adopters and states that adopted SIPOs at least X days (5? 7?) after Kentucky’s order. I concede that finding credible donors for some of the other policies may be difficult. For example, donors for the “public school closure” policy evaluation would have to rely on states that closed schools late given that these closures were nearly universal. Local policies create another complication. But a synthetic control approach seems like the most obvious research design to employ first. Relatedly, the authors could estimate a difference-in-difference (or event study) model using a sample consisting of Kentucky (the one treatment state) and the donor states. If synthetic controls are not an option to study Kentucky’s policies – perhaps because the donor pool is too limited and not credible — this could be explained in a lengthy footnote so readers know this was considered and rejected.

A second advantage of pursuing an alternative estimation strategy (if it worked out!) is that it would make the value-added of the study larger. It would avoid the problem of this paper being seen as a “small tinkering” of the Health Affairs paper.

Response: We concede that this could be a common point of confusion and have added a paragraph on pp. 16-17 to address it:

It is important to note that, while Kentucky is our focal state for the purpose of illustration, the identifying variation in our policy variables and interaction terms comes from the entire South and Midwest. This is in contrast to, for instance, a synthetic control analysis in which Kentucky is the treated state and a composite of other similar states serves as the control group. There are two main reasons for this. First, every state in the U.S. enacted multiple social distancing restrictions – and in most cases all four of the ones we consider. The identifying variation therefore comes from the timing of treatment; there is no true control group. Comparative case study methods such as synthetic control are inherently poorly suited for situations where multiple treatments were enacted at multiple points in time in multiple places. Second, utilizing identifying variation from multiple regions containing dozens of states gives the results more generalizability beyond Kentucky. While practical considerations prevent us from showing the effects of several policies at several points in time in the more than 2,000 counties in our sample, state or local policymakers interested in using our results to simulate impacts in their jurisdiction could easily do so. 

3. Is the outcome in this paper the same as the outcome in the Health Affairs paper? In the HA paper, the authors write:

“The daily exponential growth rate was calculated as the natural log of cumulative daily COVID-19 cases minus the log of cumulative daily COVID-19 cases on the prior day.” (p. 1238)

In this paper, the outcome is described as:

“We use these data to compute each county’s daily exponential growth rate in confirmed COVID-19 cases, which is equal to the natural log of daily COVID-19 cases minus the log of daily COVID-19 cases on the prior day.” (p. 6)

Why the change from growth in cumulative daily cases to growth in daily cases? A case can be made for looking at either margin (growth in day to day cumulative cases vs day to day daily cases)…both are interesting. Readers should just understand the comparability or non-comparability of the estimates across these two papers. Or maybe it was just a typo?

Response: Thanks for catching this. It was simply a typo – the measures are the same. We have added the word “cumulative” before the two instances of “daily COVID-19 cases” in the sentence in question to make it clear. 

4. The authors might mention information as an important channel through which some of these policies could affect social distancing and COVID-19 case growth. For instance, a HAH order may send important information to residents about the seriousness of the epidemic (or scare the heck out of people), which could increase stay-at-home behavior. Staying-at-home may also facilitate information gathering by watching more news reports (i.e. Daily White House Coronavirus Briefings, etc.)

Response: We agree. In addition to information, another potential channel is social pressure. Both of these are consistent with an interpretation of SIPOs as a behavioral economic nudge. Accordingly, we now write on pp. 6-7,

“SIPOs may therefore be best characterized as a “nudge” with social pressure and information about the severity of the pandemic – rather than threat of legal penalties – being the main channels through which they operate (Thaler and Sunstein, 2009). Viewed through that lens, it would not be surprising if SIPOs were less effective in areas where residents’ political views or other attributes led to skepticism of their appropriateness (Allcott, et al., 2020).” 

Note that this interpretation of SIPOs nicely motivates our new analysis showing that they have little to no effect in predominantly white, conservative, working-class counties. It makes sense that the signaling value of and social pressure from SIPOs would be relatively weak in such places. We now discuss the results accordingly in the conclusion.

5. Finally, the authors claim that in the absence of social distancing policies, COVID-19 cases in Kentucky would be an order of magnitude (11 to 12 times) higher. This a is a very large number and one that is likely to draw the attention of those skeptical of large COVID case effects of SIPOs. As the authors know, there is a recent set of studies (Cronin and Evans 2020; Goolsbee and Syverson 2020; Gupta et al. 2020; Sears et al. 2020) that emphasize that most of the variation in social distancing behavior comes from “private responses” to information/risk/beliefs rather than responses to SIPOs. (Still, plenty of credible evidence shows that SIPOs have an effect over and above these private responses.) However, given this set of papers, the authors might want to frame their effect sizes in terms of what we know about (i) social distancing elasticities with respect to the policies they study, and (ii) plausible COVID-19 case elasticities with respect to social distancing. This comment is NOT meant as an instruction to reconcile this paper’s findings with the above set of papers. Rather, the comment is designed to help the authors frame their magnitudes in the context of this growing literature.

Response: We have added the following paragraph in the conclusion on pages 21-22:

On the other hand, one might wonder how our results based on COVID-19 cases reconcile with some studies, such as Gupta et al. (2020) and Goolsbee and Syverson 

(2021), that suggest reductions in mobility largely occurred prior to SIPOs. First, measures of human mobility from cell-phone data are imperfect proxies of social distancing behaviors. The most common measure used from SafeGraph data – whether an individual stays within a “home” area for the entire day – does not capture the precautions one may take when interacting with other individuals, and becomes a worse proxy for potential interactions as population density increases. Second, as emphasized in Friedson et al. (Forthcoming), small government-induced changes in mobility can yield large changes in public health, especially when the disease is growing exponentially. Finally, the marginal compliers who voluntarily respond to initial information events about the dangers of the virus may be different than those who comply with government mandates. In practice, one likely scenario is that the reproduction rate varies with an individual’s characteristics, and those whose compliance was induced by mandates may have had more social interactions to begin with.

In terms of framing the magnitudes, we decided to simply drop the counterfactual claim in question about how much greater the number of COVID-19 cases in Kentucky would be without the restrictions. Several months have now passed since the end of our sample period, and the actual number of COVID-19 cases in Kentucky is now an order of magnitude greater than even our counterfactual estimates. In light of this, we are no longer comfortable with the strong assumptions (i.e. cases averted versus cases delayed) that would be needed to make counterfactual claims several months after most of the interventions ended. We now discuss magnitudes purely in terms of percentage point changes in the daily growth rate of cumulative cases, which should avoid some of the concerns you raised about plausibility of the bottom-line numbers.

---

## [Editor Report · Decision Letter 1]

1 Apr 2021

Chance Elections, Social Distancing Restrictions, and Kentucky’s Early COVID-19 Experience

PONE-D-20-12500R1

Dear Dr. Courtemanche,

We’re pleased to inform you that your manuscript has been judged scientifically suitable for publication and will be formally accepted for publication once it meets all outstanding technical requirements.

Kind regards,

Valerio Capraro

Academic Editor

PLOS ONE

Additional Editor Comments (optional):

I was invited to handle this paper after the first round of revision. Reading the reviews and the decision letter of the previous editor, it seems to me that the main problem with this manuscript was the lack of a clear and distinct contribution over the authors' previous paper on the topic. In the revised version of the paper, the authors address this issue in detail. They also address the other issues raised by the reviewers. I have to say that I tried to invite the reviewers to review the revised version of the paper, but none of them agreed to review (these are difficult times and it is becoming more and more difficult to find reviewers). My feeling, however, is that the authors have addressed the issues and there is no need to search for a third reviewer and start the review process again, especially because there was no critical issues to be solved. Therefore, I think the paper can be accepted as is.
---

## [Editor Report · Acceptance letter]

23 Jun 2021

PONE-D-20-12500R1 

Chance Elections, Social Distancing Restrictions, and Kentucky’s Early COVID-19 Experience 

Dear Dr. Courtemanche:

I'm pleased to inform you that your manuscript has been deemed suitable for publication in PLOS ONE. Congratulations! Your manuscript is now with our production department. 

Kind regards, 

on behalf of

Dr. Valerio Capraro 

Academic Editor

PLOS ONE